# Thermal Stability and Matrix Binding of Citrinin in the Thermal Processing of Starch-Rich Foods

**DOI:** 10.3390/toxins17020086

**Published:** 2025-02-13

**Authors:** Lea Brückner, Florian Neuendorff, Katharina Hadenfeldt, Matthias Behrens, Benedikt Cramer, Hans-Ulrich Humpf

**Affiliations:** Institute of Food Chemistry, University of Münster, 48149 Münster, Germany; lbrueckn@uni-muenster.de (L.B.); mattbehrens@uni-muenster.de (M.B.)

**Keywords:** citrinin, mycotoxin, food, degradation, matrix binding, modified mycotoxin

## Abstract

Citrinin (CIT) is a nephrotoxic mycotoxin commonly found in a broad range of foods, including cereals, spices, nuts, or *Monascus* fermentation products. Analyses have shown that CIT is present in processed foods in significantly lower concentrations than in unprocessed materials. Modified forms of CIT arising during food processing may provide an explanation for the discrepancy. This study deals with the thermal stability of CIT and the formation of reaction products of CIT with carbohydrates, followed by toxicological evaluations using cell culture models. HPLC-HRMS degradation curves of CIT heated in different matrix model systems were recorded, and the formation of decarboxycitrinin (DCIT), the main degradation product, was quantified. Additionally, chemical structures of reaction products of CIT with carbohydrates were tentatively identified using MS/MS spectra and stable isotope labelling. Subsequently, the degradation of CIT during biscuit baking was studied, and carbohydrate-bound forms of CIT were detected after enzymatic starch digestion. The formation of DCIT could explain the majority of CIT degradation, but, depending on the process, covalent binding to carbohydrates can also be highly relevant. Cytotoxicity of DCIT in IHKE-cells was found to be lower compared to CIT, while the toxicity as well as the intestinal metabolism of carbohydrate-bound CIT was not evaluated.

## 1. Introduction

The mycotoxin citrinin (CIT, Figure 1) is primarily found in cereals and other plant-based products such as spices, beans, fruit, nuts, or olives [1,2,3,4,5,6]. In 2017, occurrence data was largely extended by a comprehensive study in which more than 1000 food samples were analyzed for CIT. One result of this study was that cereals, and cereal products in particular, were contaminated with CIT [2]. Cereal products, especially baked goods, are important staple foods worldwide [7,8]. Due to their consumption quantities, cereal products also have the greatest impact as a source of mycotoxins when comparing different foods, making them a key factor in CIT exposure [8].

CIT exists in two tautomeric forms, with a 3:2 equilibrium between the *p*- and *o*-tautomers in the solid state at room temperature (Figure 1) [9,10]. CIT is produced by different genera of *Penicillium*, *Aspergillus*, and *Monascus*, and is of toxic relevance primarily due to its nephrotoxicity, which has been demonstrated in vivo and in vitro [11,12,13].

Cereal manufacturing can include different types of physical or chemical processing such as sorting, sieving, washing, steeping, milling, and alkaline or heat treatment. Each step can affect the mycotoxin concentration in the final product. Processing steps that apply thermal energy or chemical treatment may also have an impact on the chemical structure of the toxin, leading to chemical modification of the toxin itself or its reaction with food constituents to generate matrix-bound forms [8,14,15,16,17,18].

In a broad-based study on the occurrence of CIT in food, it was observed that the CIT concentrations in processed cereal products (1.0–5.7 μg/kg, mean value of all samples above the limit of quantification: 2.2 μg/kg) were, on average, tenfold lower than in unprocessed cereal products (1.1–155 μg/kg, mean value of all samples above the limit of quantification: 20.9 μg/kg) [2]. This large discrepancy cannot be fully explained by CIT removal due to physical processes but suggests that chemical modifications of CIT occur during manufacturing, leading to various forms of modified CIT that cannot be detected using conventional analytical methods.

Modified forms of mycotoxins pose a fundamental problem in the analysis of mycotoxins, as they can contribute to the underestimation of mycotoxins in food and feed [15,19,20,21,22,23]. The terminology related to modified mycotoxins has not been consistent in the past, with synonyms such as masked, conjugated, hidden, or bound forms not being used systematically [14,18,24,25]. This article will adopt the common definition of modified mycotoxins according to Rychlik et al. [14], which is also used by the European Food Safety Authority (EFSA) [26]. A distinction is made between chemically modified mycotoxins, which can arise in thermal and non-thermal food processing, and matrix-associated mycotoxins, where a distinction is made between covalently and non-covalently bound forms [14].

There are examples of chemically modified mycotoxins that can be either more or less toxic than the parent compound. However, matrix-associated, covalently bound mycotoxins pose a particular problem. When bound to macromolecules, these compounds lose their direct bioavailability. However, the parent mycotoxin or small molecule derivatives thereof may be released during enzymatic digestion in the intestinal system, as was shown for the glucosidically bound T-2 and HT-2 toxins [27]. From an analytical point of view, matrix-associated mycotoxins are also challenging, as they must first be liberated by chemical or enzymatic treatment before analysis [15,16,18,22,28,29].

In cereal products, reactions of mycotoxins with carbohydrates generating matrix-associated mycotoxins are particularly important, as the proportion of carbohydrates is manyfold higher than that of proteins. Examples of such reactions are the formation of carbohydrate esters of fumonisin B_1_ and ochratoxin A via their free carboxylic acid functional group. In the case of fumonisin B_1_, the possible matrix binding through esterification was demonstrated in model experiments. In the case of ochratoxin A, this was also observed during coffee roasting in addition to model experiments [30,31]. Binding of fumonisin B_1_ to glucose via the amino group of the toxin is also possible [32]. Additionally, matrix-associated forms covalently bound to carbohydrates of T-2 toxin have been detected in both model experiments and baked biscuits. Unlike in the case of fumonisin B_1_ and ochratoxin A, however, binding to carbohydrates does not occur via a carboxyl or amine group since the T-2 toxin lacks these functional groups. Instead, binding happens via addition to the double bond, which also irreversibly alters the chemical structure of the toxin [33,34]. Moreover, oligoglycosides of deoxynivalenol have been described for cereal-based products [35]. In addition, covalently bound forms of mycotoxins can be formed via reactions with the thiol, amine, or hydroxy groups of proteins [36,37]. Due to the structural similarity between CIT and OTA (both molecules possess a carboxyl group), reactions with hydroxy groups of carbohydrates and, thus, the formation of matrix-bound forms are also conceivable for CIT.

Studies investigating the degradation of CIT are limited and, so far, solely performed with the pure compound. Model experiments heating CIT in solution at different pH values and temperatures showed the formation of dimerization products (dicitrinin A–D) and degradation products such as CIT H1, CIT H2, phenol A, phenol A acid, and decarboxycitrinin (DCIT). Studies on the cytotoxicity of these compounds revealed that CIT H1 exhibited higher cytotoxicity in HeLa cells than CIT, while CIT H2 was less cytotoxic. Dicitrinin A was described as moderately active in the NS-1 cytotoxicity assay [38,39,40]. However, most of the degradation products described above were only observed in model experiments without other food constituents as matrix. In 2012, the EFSA stated that there is insufficient data available on the occurrence of CIT in food and feed and that the possibility of matrix-associated CIT caused by food or feed processing has not yet been investigated [1]. Recently, we could, for the first time, show that CIT reacts with amino compounds, such as gluten, during thermal processing, confirming that this mycotoxin can react with macromolecules [36]. The aim of this study was to investigate whether citrinin can also react with carbohydrates. Starting from experiments with model compounds, we studied the formation of starch-bound CIT and monitored its formation during biscuit baking as thermal food processing.

## 2. Results

### 2.1. Reactions of CIT with Different Carbohydrates

Different matrix model compounds were used to investigate reactions between CIT and carbohydrates. Thereby, methyl-α-d-glucopyranoside was used as a model compound for starch since the anomeric C-atom is protected, whereas all other hydroxy groups are available for potential reactions with CIT comparable with starch. Reducing sugars were represented by α-d-glucose, while non-reducing sugars and disaccharides were represented by d-sucrose. In a control experiment, CIT was heated under the same conditions without the addition of a model compound. Additionally, the matrix model compounds were heated without CIT to rule out potential interferences with formed reaction products of CIT. Detectable signals from the heated pure model compounds eluted within the first two minutes, confirming that interferences could be excluded.

Pure, water-free CIT is stable at temperatures up to 120 °C, as shown in Figure 2. However, the addition of water decreases its stability, resulting in more than 60% CIT reduction after 10 min at 100 °C (Appendix A). This also results in the formation of several degradation products, including DCIT, phenol A acid, dicitrinin A, dicitrinin C, citrinin H1, and two structurally unknown degradation products with mass-to-charge ratios (*m*/*z*) of *m*/*z* 395.1856 ([M+H]^+^, C_24_H_26_O_5_) and *m*/*z* 425.1960 ([M+H]^+^, C_25_H_28_O_6_), which are present in relatively high amounts based on signal intensity [36].

The addition of model compounds also strongly affects the CIT stability, resulting in a faster degradation of CIT compared to the control (Figure 2).

A direct comparison of HPLC-HRMS data for heated pure CIT and CIT heated together with α-d-glucose is shown in Figure 3 and revealed the formation of new peaks at a retention time window between 4.5–5.7 min in the presence of the reducing sugar. Three main peaks (see enlargement in Figure 3) with *m*/*z* 351.1445 ([M+H]^+^, C_18_H_22_O_7_, calculated *m*/*z* 351.1438, error: 2.0 ppm) were detected and are referred to as reaction products **B** in the following. In addition, smaller isobaric peaks with *m*/*z* 369.1540 ([M+H]^+^, C_18_H_24_O_8_, calculated *m*/*z* 369.1544, error: 1.0 ppm) were observed at the same retention times (see enlargement in Figure 3), which are referred to as reaction products **A** in the following. Reaction products **A** and **B** show a mass difference of 18.0106 Da, corresponding to one molecule of H_2_O.

Additional confirmation that the reaction products **A** and **B** are formed by reaction of CIT with α-d-glucose was obtained by repeating the experiments with stable isotope-labeled reactants. First, ^13^C_3_-labeled CIT and α-d-glucose were heated together, and second, CIT was heated with ^13^C_6_-labeled α-d-glucose. The observed reaction products of both experiments had the same retention times as the non-labeled products. For the reaction with ^13^C_3_-labeled CIT, *m*/*z* 354.15454 (+3.0100 Da compared to **B**) and *m*/*z* 372.1641 (+3.0101 Da compared to **A**) were determined, while for the reaction product with ^13^C_6_-α-d-glucose, *m*/*z* 357.1621 (+6.0176 Da compared to **B**) and *m*/*z* 375.1725 (+6.0181 Da compared to **A**) were observed. These findings unequivocally confirm that one molecule of CIT and one molecule of α-d-glucose react with each other while maintaining all labeled C-atoms.

The comparison of the product ion spectra of *m*/*z* 351.1445 (product **B**) with those of *m*/*z* 354.1544 (^13^C_3_-CIT reacted with α-D-glucose) and of *m*/*z* 357.1623 (CIT reacted with ^13^C_6_-α-D-glucose) provided further insights into the molecular structure (Figure 4).

The fragment ion *m*/*z* 303.1212 results from the loss of CH_2_O originating from glucose, as well as the loss of one molecule of water. This is confirmed in the experiment with ^13^C_6_-α-d-glucose and CIT, where the corresponding product ion has *m*/*z* 308.1376, which is only 5.0164 Da, i.e., the mass difference between ^13^C_5_ and ^12^C_5_. A similar fragment, *m*/*z* 285.1112, shows a further water loss compared to *m*/*z* 303.1212 of the unlabeled reaction product. The product ion spectra of the isotope-labeled compounds show corresponding fragments with *m*/*z* 288.1229 and 290.1275. The CH_2_O loss is a characteristic feature of glucose and other sugars [41,42,43] and is identified as the X-fragment according to the Domon−Costello nomenclature [41,44]. The fragment ion *m*/*z* 259.1314 also originates from the sugar fragmentation, as confirmed by the ^13^C_6_-α-d-glucose experiments. The fragment ion *m*/*z* 219.1001 results from the cleavage of a large part of the sugar (-C_5_H_8_O_4_). This is further confirmed by the experiment with ^13^C_6_-α-d-glucose, where all five cleaved C-atoms were isotope-labeled, resulting in *m*/*z* 220.1036 with only one ^13^C remaining. It is assumed that the cleavage occurs at C-6 of the sugar in an analogous way, as described in [42]. The *m*/*z* 207.1003 fragment corresponds to a CIT fragment in which the entire sugar moiety has been cleaved off (-C_6_H_8_O_4_). This is confirmed by the experiment with ^13^C_3_-labeled CIT, where the resulting fragment, *m*/*z* 210.1121, is exactly 3.0118 Da heavier. Additionally, the experiment with ^13^C_6_-α-d-glucose showed that all ^13^C-labeled atoms are cleaved off. This sugar cleavage is also described as a typical fragmentation reaction [41]. In summary, the product ion spectra of **B** thus show the CIT fragment (*m*/*z* 207.1015), along with the neutral loss of α-d-glucose minus two water molecules (C_6_H_8_O_4_) and characteristic glucose fragmentations [41,42,43]. The recorded product ion spectra of **A** (Appendix A) are comparable to those of **B**, showing the CIT fragment (*m*/*z* 207.1013) and the same characteristic sugar fragmentations. No differences in the fragment spectra could be observed between the different isobaric peaks of **A** and **B**.

Structurally, CIT is related to ochratoxin A: Both mycotoxins are aromatic polyketides and possess a reactive carboxylic acid group. As already described previously [31], the carboxylic acid group of ochratoxin A can react with the hydroxy groups of α-d-glucose (preferably with the primary hydroxy group at C-6 of α-d-glucose) to form the corresponding esters. Accordingly, a reaction between CIT and α-d-glucose would theoretically result in a reaction product with the mass of 412.1364 Da for C_19_H_24_O_10_ and *m*/*z* 413.1442 ([M+H]^+^) or *m*/*z* 435.1262 ([M+Na]^+^) in positive ionization mode. However, this could not be observed or detected only in traces (the maximum peak area for *m*/*z* 435.1262 was 2.0E+04 counts, and [M+H]^+^ was not observed at all after a 10 min heating time at 160 °C). Instead, as described above, mainly *m*/*z* 351.1445 ([M+H]^+^, C_18_H_22_O_7_) (reaction products **B**) was visible, with a maximum peak area of 2.4E+06 counts. When comparing the sum formulas of the theoretically expected reaction product (C_19_H_24_O_10_) and the observed reaction products (C_18_H_22_O_7_), it is clear that a loss of CO_2_ and a further loss of water must have taken place during the reaction. The verification experiments with ^13^C_3_-labeled CIT and ^13^C_6_-labeled α-d-glucose, along with the observed reaction products, showed that a loss of CO_2_ could not have resulted from the α-d-glucose itself. Furthermore, none of the three labeled ^13^C-atoms of the CIT depicted in Figure 1 were cleaved off. The most likely explanation is that the carboxyl group of the CIT was cleaved off, but at the same time, this group would be involved in an ester-forming reaction with one of the hydroxy groups of α-d-glucose.

Indeed, we propose that the ester formation is the first step of the reaction of CIT with α-d-glucose, leading to the formation of a β-keto ester. This process results in a strong positive partial charge of C-6 of the α-d-glucose molecule. The next step presumably involves cyclization and the elimination of CO_2_ from the molecule. The elimination of another water molecule to form **B** can occur either from the CIT or through the loss of a hydroxy group from the sugar, with the latter seeming more likely. Reaction products **A** are compounds in which only CO_2_ is eliminated after ester formation, without the additional loss of a water molecule compared to **B**. The formation of different isomers of **A** (and **B**) can arise from the reaction of different hydroxy groups of the sugar with CIT during the ester formation, with the primary OH-group at C-6 probably being the most favored (structure in Figure 5). In addition, the loss of different OH-groups in the reaction step leading to **B** can result in an even higher number of isomers. Therefore, the postulated reaction pathway in Figure 5 illustrates only one possible example, in which CIT reacts with the primary hydroxy group at C-6 of the α-d-glucose. Unfortunately, structural elucidation of the resulting reaction products by NMR and confirmation of the postulated reaction pathways was not possible due to low yields and the complexity of the resulting reaction products.

When CIT is heated with d-sucrose, **B** in particular is detectable, similar to the experiment with α-d-glucose. In addition, small amounts of *m*/*z* 513.1976 ([M+H]^+^, C_24_H_32_O_12_) are formed. The reaction pathway is presumed to be identical to the reaction pathway shown above (Figure 5). The fact that mainly **B** is detected indicates that a glucose molecule is either already split off during heating or that the reaction products *m*/*z* 513.1976 are unstable in the ESI source. Additional confirmation of the reaction between CIT and α-d-sucrose was again obtained by repeating the experiments with stable isotope-labeled ^13^C_3_-CIT, and, as in the case for α-d-glucose, the reaction was confirmed.

Methyl-α-d-glucopyranoside was used as a model compound for starch. It is notable that the CIT degradation with the addition of this model compound was overall stronger than for the other model compounds investigated (see Figure 2). However, reaction products could not be identified with the HPLC-DAD-ESI-QTOF method used, or only in traces. For this reason, CIT was heated again with starch itself instead of the model compound. Since the direct detection of reaction products from CIT and macromolecules is not possible due to the high molecular weight, the first goal was to develop a method in which monomer or oligomer subunits of the macromolecule bound to the mycotoxin can be detected. For this purpose, the samples were enzymatically treated with α-amylase. The experiments were carried out in triplicate and, as a control, starch was heated without the addition of CIT and then enzymatically digested as well. Possible reaction products were identified using a software-based approach via MetaboScape (Bruker), similar to the approach described in Brückner et al. [36].

As a first step, the data was processed using an intensity threshold of 10.000 cps for MS^1^
*m*/*z* signals to generate feature lists. For each feature, the respective *m*/*z* is combined with the retention time, peak signal intensity, possible adduct ions, isotope pattern, and, if present, fragment ions. For statistical comparison of the test and control groups, univariate analyses were performed using fold-change analysis in combination with *t*-test analysis. The results are displayed in a volcano plot (Figure 6) to visualize differences between the test and the control group.

The following filter criteria were used to decide whether a feature was considered as a reaction product of CIT and starch or not:The feature had to have a fold change >2, and the significance level was set at 90% (*p*-value < 0.1);The exact mass had to be assigned to a reasonable sum formula, consisting of C-, O-, and H-atoms (error to the calculated sum formula was not allowed to be greater than 5 ppm) and a *m*/*z* ≥ 300;The detected feature had to have a retention time shorter than the retention time of CIT (approx. 8.3 min) as the polarity of these reaction products is expected to increase, resulting in less retention than CIT on a reversed-phase column;Features identified in experiments where CIT was heated without additives were excluded.

A total of 104 features were identified in the test group, of which 17 were found to be statistically significant (fold change > 2, *p*-value < 0.1) (Figure 6). After applying the filter criteria, five of these features remained as possible reaction products of CIT with starch (Table 1).

Three features were already known from the heating experiments with α-d-glucose and d-sucrose: two isomers of C_18_H_24_O_8_ (*m*/*z* 369.1548/ 369.1546, retention time of 4.95 min and 5.16 min, reaction products **A**), as well as *m*/*z* 351.1440 (**B**). The remaining two features showed *m*/*z* 531.2080 ([M+H]^+^, C_24_H_34_O_13_) and *m*/*z* 693.2749 ([M+H]^+^, C_30_H_44_O_18_), which correspond to CIT bound to a di- and trisaccharide, respectively. In these cases, the starch was most probably not completely digested to the glucose monomer, so that CIT is bound, for instance, to an α-d-maltose (*m*/*z* 531.2080) or α-d-maltotriose (*m*/*z* 693.2749). Both products were formed by the loss of one CO_2_ and one H_2_O-molecule, as described for **A**. In contrast to the model experiments with α-d-glucose and d-sucrose, the intensities of the observed reaction products **B** were lower than for **A**. This suggests that the additional water cleavage observed in the model experiment with mono- or disaccharides is of less relevance for the starch binding of CIT.

### 2.2. Reactions and Degradation of CIT in Starch-Rich Baking Products

The model experiments showed that CIT can react with sugars as well as larger macromolecules such as starch. Thus, the next step was to investigate the reactions of CIT under food-processing conditions such as baking. To exclude any chemical changes in the mycotoxin due to fermentation or the effect of baking powder, a simple non-yeast biscuit recipe, based on Kuchenbuch et al., was used to investigate the degradation of CIT under realistic food processing conditions [34]. The biscuit dough was made with wheat flour, water, coconut fat, and sugar.

To investigate the influence of different ingredients on the thermal degradation of CIT, the following ingredients were varied:Moisture content: low, high, and medium moisture (doughs A, B, and C);Sugar: non-reducing sugar (sucrose, dough C) and reducing sugar (glucose, dough D);Flour types: type 405, type 550, and whole grain wheat flour (doughs C, E, and F).The recipes of the used dough A–F are summarized in Table 2.

One part of the dough was spiked with CIT (776.2–782.5 µg CIT/kg dry mass dough, compare Figure 7), while the other part remained unspiked. All doughs A–F were baked using two different baking methods:High-temperature conditions (220 °C for 10 min) with a reduced baking time;Low-temperature conditions (180 °C for 20 min) with an extended baking time.

These two baking conditions were chosen to produce biscuits with a similar degree of browning. The temperature inside the biscuits was monitored during the baking. Representative temperature curves are shown in Figure 6 for dough C for both baking conditions.

The maximum core temperature of the biscuits hardly exceeded 100 °C under either baking condition, indicating that, until the end of the baking process, residual moisture remained. Initially, dough C had a water content of 25%, with a residual moisture content of 19% after low-temperature baking and 22% after high-temperature baking. For the high-temperature conditions, the core temperature reached 100 °C after 6.8 min, slightly earlier than under low-temperature conditions, where 13.1 min were needed to reach 100 °C. However, the core temperature of 100 °C was maintained for 6.9 min under low-temperature conditions, which is more than twice as long as under high-temperature conditions (3.2 min). The samples were analyzed both for extractable and starch-bound forms of CIT using two different procedures (discussed below under Section 2.3 and Section 2.4).

### 2.3. Extractable CIT Reaction Products Formed During Biscuit-Making

To quantify CIT and the known degradation product DCIT, as well as to screen for other directly extractable, free degradation products, biscuit samples were extracted and directly analyzed by HPLC-MS/MS. In addition to CIT, the degradation product DCIT was present in all samples, while no signals corresponding to other degradation products were observed. The determined levels of CIT and DCIT, reported as CIT equivalents and normalized to the dry mass of the biscuit, are depicted in Figure 8 and Appendix A.

It is noteworthy that the low-temperature conditions resulted in slightly higher CIT-degradation rates, with residual CIT levels ranging from 68 to 86% in the baked doughs A–F. In contrast, under high-temperature baking conditions, 74 to 97% of CIT could still be detected after baking of doughs A–F (Table 2). This can be explained by the fact that the core temperature of the biscuits did not significantly exceed 100 °C, but that this temperature was reached over a longer period with the low-temperature baking conditions. The higher degradation rate of CIT observed in the model experiments when heating CIT with α-d-glucose compared to d-sucrose could not be confirmed in the biscuit baking experiment (comparison of dough C and D). However, a stronger CIT reduction by −30% (low-temperature conditions) and −23% (high temperature conditions) was recorded for the baking of dough F compared to all other doughs. Dough F was the only dough prepared with whole grain wheat flour (Table 2).

DCIT was formed in comparable amounts in all doughs. Under the low-temperature conditions, the quantities formed were slightly higher (3–12%) than in the high-temperature conditions (4–7%). The smallest level of DCIT formed was observed in dough D under the low-temperature conditions (3%).

### 2.4. Matrix Binding of CIT During Biscuit-Making

To identify starch-bound CIT that may be present in biscuits due to thermal processing, the starch was enzymatically digested. Only the samples from the low-temperature conditions were subjected to starch digestion, as these exhibited higher overall degradation rates. To improve the yield of the enzymatic hydrolysis, the samples were treated with amyloglucosidase from *Aspergillus niger* to hydrolyze exo-α-d-(1-4)- and branch point α-d-(1-6)-linkages after α-amylase digestion [45,46]. Details on the optimization of this process can be found in the supplement. After enzymatic digest, the samples were diluted and subjected to HPLC-DAD-ESI-QTOF and, for enhanced sensitivity, to HPLC-MS/MS analysis.

For this purpose, a series of different MRM transitions were integrated into an HPLC-MS/MS method using the mass spectra generated by HPLC-DAD-ESI-QTOF, as well as sample material from the model experiment heating CIT with starch.

To detect the formed amount of the reaction products **A**, two MRM transitions were selected, and signal intensities were recorded. Unfortunately, quantification of the reaction products was not possible due to missing analytical standards. To provide a point of reference, the calibration for CIT was used for comparison, although it is assumed that the ionization and fragmentation efficiency for **A** is lower. It was confirmed that **A** was formed during biscuit-baking. However, no reaction products **B** were detected in these samples.

Figure 8 summarizes the recovered CIT level, the formed DCIT, and the semi-quantitatively determined amount of matrix-associated CIT covalently bound to starch, each converted into CIT equivalents. Based on the recorded signal intensities, the amount of matrix-associated CIT detected in the baked biscuits seems to be lower compared to the DCIT formed and the CIT recovered after baking. Differences in the amount of matrix-associated CIT formed during biscuit-making are minimal for all doughs tested. For the biscuit doughs A, B, C, and E, it can be assumed that a large part of the CIT spiked before baking was recovered in the form of DCIT, CIT covalently bound to starch (**A**), or unaltered CIT. In dough D (which used α-d-glucose), a similar amount of CIT was degraded compared to the doughs A, B, C, and E, but less DCIT was formed. This resulted in a gap compared to the spiked CIT, with approximately 82 µg of the spiked CIT missing.

Similarly, in dough F (which used whole grain wheat flour), where CIT degradation was highest, the amount of CIT covalently bound to starch was not higher than in the other biscuits, and thus cannot explain the large gap to the previously spiked CIT (around 182 µg CIT are missing).

As we have recently shown, amino compounds are further reaction partners for CIT during thermal processes [36]. In addition to the reaction products of CIT with starch, it should also be evaluated whether matrix-associated CIT covalently bound to proteins is formed in the baking samples as well. A similar approach to the previously described model experiment was used, in which CIT was heated together with gluten. The protein fraction of the biscuit samples was enzymatically hydrolyzed using pronase (from *Streptomyces griseus*) to allow subsequent detection of the reaction products by mass spectrometry [36]. To improve the enzymatic digestion of the proteins, as much of the starch as possible should be removed in advance. For this purpose, the samples were first incubated with α-amylase and the supernatant including the soluble sugars was removed. The proteins were then digested using pronase and the samples were checked for the recently published reaction products of CIT and gluten using HPLC-DAD-ESI-QTOF.

However, none of the features recently described could be detected in any of the representatively selected baking samples. In addition, the theoretically expected reaction product of CIT and lysine could also not be observed.

Given that there might still be other unknown reaction products, the extracts of the enzymatically digested biscuit samples were also analyzed using an untargeted approach, similar to that used for the extractable CIT reaction products, using an HPLC-DAD-ESI-QTOF instrument. However, no further degradation products could be detected in these samples, leaving DCIT as the most abundant CIT reaction product. Consequently, the question arises whether this formation may be a detoxification process of CIT.

### 2.5. Cytotoxicity of CIT and DCIT

The toxicity of DCIT was first studied in mice by Jackson and Ciegler [47], who reported that DCIT was not toxic under the experimental conditions used. However, the substance was not fully purified or entirely elucidated prior to injection, and it is unclear how many mice were used and in what time period the study was conducted. Devi et al. were able to show that DCIT retains its antibiotic activity when transformed from CIT to DCIT (tested against strains of *Salmonella typhi*, *Escherichia coli*, *Klebsiella* sp., *Vibrio cholerae*, *Staphylococcus pyogenes,* and *Acinetobacter* sp.) [48]. This is in contrast to data published shortly afterwards, in which none of the isolated substances from model experiments (including DCIT) had significant antimicrobial activity (no further details are given about the microorganism tested) [40]. In contrast to CIT, no data has yet been published on the cytotoxicity of DCIT. However, since the main target of CIT is the kidney [1], IHKE-cells were selected as complementary cell culture model. CIT and DCIT were applied to the cells in concentrations ranging from 500 nM to 100 μM.

The data shown in Figure 9 illustrate that, overall, the cell viability of IHKE-cells was lower for CIT compared to DCIT across all tested concentrations. However, from a concentration of 2 µM, a significant decrease in cell viability (*p* ≤ 0.05) was also observed for DCIT. Therefore, it can be concluded that DCIT formed during thermal processing is less cytotoxic compared to CIT, but it is not entirely detoxified. When comparing 100 µM DCIT (69 ± 10%) and 20 µM CIT (70 ± 10%), similarly significant effects on cellular viability were observed.

## 3. Discussion

Knowledge about the CIT degradation during food or feed processing is limited and the occurrence of modified forms of CIT has not yet been adequately studied [1]. As we have recently shown, CIT is able to react with various amino compounds under food processing conditions, leading to protein-bound forms of CIT [36]. To extend these studies, we investigated the reactions of CIT with different carbohydrates during thermal food processing, using, in the first step, different matrix model compounds representing different carbohydrates. The suitability of these model compounds to mimic food constituents was demonstrated in previous studies that investigated the thermal degradation of the mycotoxins fumonisin B_1_, deoxynivalenol, nivalenol, ochratoxin A, T-2, and HT-2 toxin in food (see [30,31,32,33,36] and literature cited there).

In summary, our data demonstrate that CIT is a reactive mycotoxin that begins to degrade under comparatively mild thermal processing conditions, starting at 100 °C. The observed higher degradation rate in the presence of water during heating is in good agreement with findings from the literature [49]. In addition, thermal degradation is more pronounced in the presence of food matrices. The results from the matrix model experiments provide clear evidence that, in addition to amino compounds (e.g., proteins), sugars and other carbohydrates such as starch are also possible reaction partners for CIT during thermal food processing. MS/MS spectra and stable isotope labelling experiments were used to further characterize the formed reaction products **A** and **B**, which are assumed to be products of the decarboxylation of a β-keto ester. In the literature, the decarboxylation of β-keto acids is described upon heating, which supports the hypothesis that these products arise from such a reaction [50]. The formation of numerous isomers can be attributed to the five hydroxy groups of glucose, each of which could act as potential reaction partner for CIT. A similar phenomenon has also been observed in the reactions of T-2 toxin with carbohydrates [33]. The further loss of different hydroxy groups (leading to reaction products **B**) results in an even greater number of potentially formed isomers. However, due to low yields and the complexity of the mixture of reaction products, further structural elucidation, for example, by NMR analysis, was not possible.

In a next step, to verify the results of the model reactions of CIT with carbohydrates under realistic food processing conditions, we performed baking experiments commonly used in the food industry. Since the proportion of starch is usually significantly higher than the amount of protein in cereal and cereal products [51], i.e., foods that are often contaminated with CIT, a reaction of CIT with carbohydrates seems most likely.

Two different baking conditions, representative of typical industrial baking processes [34], were compared. These conditions varied in baking time and temperature but were intended to produce a similar baking result (i.e., a similar degree of browning). A higher degradation of CIT was observed when longer baking times at lower temperatures were used.

Overall, between 68–97% of the added CIT could be detected in the samples after biscuit-baking, demonstrating that CIT partly reacts during this process. The largest effects on CIT degradation were observed when baking biscuits from a whole grain wheat flour dough (dough F). However, the specific effect of whole grain wheat flour on the thermal degradation of CIT can only be speculated. Compared to wheat flour types 405 or 550, whole grain wheat flour is characterized by a slightly higher protein content (1.5% higher), a slightly higher fat content (1.1% higher), a higher proportion of dietary fiber (7.6% higher), and a slightly higher proportion of minerals (1.1% higher) [51]. For amino compounds such as proteins, we have already shown recently that they have an influence on the degradation of CIT [36]. The available cellulose and phenolic macromolecules may also react with CIT in a manner similar to starch. Minerals (especially heavy metals such as iron, copper, or cobalt) can be involved in oxidation processes and act as reaction catalysts in (thermal) food processes [52,53,54]. This could provide plausible explanation for the lowest CIT recovery observed in dough F.

The formation of DCIT seems to explain the majority of CIT degradation, with starch-bound CIT (reaction products **A**) potentially playing a decisive role depending on the process conditions and ingredients used. In this context, it has to be mentioned that no standards for the quantification of **A** were available, and thus, the levels shown in Figure 8 are for orientational purposes only. Depending on the efficiency of the enzymatic release of the reaction products from the starch matrix and, even more important, the ionizability of the reaction products in the mass spectrometer, the quantities formed may be significantly higher. For dough F, the identified starch-bound CIT could not explain the smaller proportion of recovered CIT compared to all other doughs examined. In fact, the amount of CIT covalently bound to starch formed in this dough is the lowest compared to all other doughs. One possible explanation might be the presence of additional cellulose and other dietary fiber, which may react with CIT in a manner similar to starch. In addition, the relevance of matrix-bound CIT to pectin in apples has already been discussed [55].

Reaction products **B** could not be analyzed in the baked biscuits. Since the CIT concentrations used in the baking experiments were lower than in the previously conducted model experiments, the total amount of reaction products **A** formed during the baking experiments was also considerably lower. It is assumed that small amounts of reaction products **B** may have been formed, but these amounts were below the detection limit.

Protein-bound forms of CIT were not detected in any of the doughs. On the one hand, it is conceivable that the amount of CIT covalently bound to proteins is low due to the rather low protein content in the baked goods, making it undetectable with the QTOF detector used, which has comparably low sensitivity. In addition, it must also be emphasized that the recently presented reaction products of CIT with gluten were derived from a model experiment in which gluten was selected as a representative main component of the protein fraction in wheat [56] and the features identified there do not necessarily represent other proteins. Therefore, it cannot be ruled out that reaction products of CIT with other amino compounds were formed. However, these could not be detected in the baked goods due to the absence of a reference list. Moreover, the influence of potential matrix interferences remains unknown.

Overall, it should be noted that CIT degradation in the biscuit-baking experiments was rather low under all conditions investigated. Kuchenbuch et al. observed higher degradation rates for T-2 and HT-2 toxin, but it is important to emphasize that significantly harsher baking conditions were used (200 °C for 30 min/ 230 °C for 20 min) [34]. However, the biscuit-baking experiments conducted in this study showed that the ingredients, the moisture level, and the baking time and temperature strongly influence the stability of CIT. Higher degradation rates might occur in other products, particularly those baked under harsher conditions. Additionally, the detected analytes demonstrate that starch-bound forms of CIT can be formed during commonly applied baking processes. In addition to the already confirmed reactions of CIT with amino compounds such as proteins during heating [36], the reactions of CIT with carbohydrates, as presented here, may offer further insight into why processed starch-rich foods generally contain less CIT than unprocessed ones.

Based on the assigned chemical structures, these covalently bound forms may be at least partially released to their bioavailable form, particularly during digestion, and thus regain toxicological relevance. In the case of CIT covalently bound to starch, however, it is important to note that the covalent binding has very likely resulted in a chemical change (Figure 5), leading most probably to different toxicological properties compared to unmodified CIT.

Our data also indicate that model experiments involving the heating of only pure CIT are not suitable for accurately predicting the degradation products that can actually arise during thermal food processing [38,39,40]. Kitabatake et al. [49] investigated the degradation of CIT under dry, semi-moist, and moist conditions in model experiments, where pure CIT was heated both with and without water but without any food matrix. Degradation products formed were later identified as CIT H1 and CIT H2 [38,39]. Other degradation products, such as the dimers dicitrinin A–D or DCIT, have been identified in model experiments where pure CIT was heated in methanol [40]. Nevertheless, under realistic food processing conditions and in the presence of a food matrix, we could not identify any of the degradation products described in literature, except for DCIT. Dimerization and other CIT-CIT couplings reported in these studies are very unlikely under typical food processing conditions due to the strong dilution with other food constituents.

Cytotoxicity of DCIT in IHKE-cells was found to be lower compared to CIT, but there is no evidence for complete detoxification. As DCIT is the main low molecular weight degradation product of CIT under realistic food processing conditions, it is recommended to quantify DCIT in processed food samples in addition to CIT. The toxicity, as well as intestinal metabolism, of carbohydrate-bound CIT could not be evaluated due to the relatively small amounts formed. However, the release of matrix-bound forms of CIT during intestinal metabolism, as well as the toxicity of the released forms of CIT, should be analyzed in future studies.

## 4. Materials and Methods

### 4.1. Chemicals and Reagents

CIT (purity ≥ 98%) was purchased from Enzo Life Sciences (Lörrach, Germany). ^13^C_3_-labeled CIT was synthesized by Bergmann et al. [57]. DCIT (purity > 98%) was isolated as described in Brückner et al. [36]. DH-CIT was purchased from AnalytiCon Discovery (Potsdam, Germany). α-d-Glucose (anhydrous analytical grade) was purchased from Serva (Heidelberg, Germany). α-d-^13^C_6_-Glucose (99%) was purchased from Deutero GmbH (Kastellaun, Germany). d-Sucrose (≥99.5%), sodium acetate trihydrate (≥99%), magnesium sulphate (≥99%), and sodium chloride (≥99%) were purchased from Carl Roth (Karlsruhe, Germany). Methyl-α-d-glucopyranoside (≥99%), starch (soluble, ACS reagent grade), and α-amylase from *Bacillus sp*. were purchased from Sigma-Aldrich (Steinheim, Germany). Acetic acid (100%), formic acid (>99%), dimethyl sulfoxide (DMSO) (≥99%), and amyloglucosidase from *Aspergillus niger* were purchased from Merck KGaA (Darmstadt, Germany). Purified water was generated using a Purelab Flex 2 system (Veolia Water Technologies, Celle, Germany). Acetonitrile (ACN) of LC-MS grade and methanol of LC-MS grade were purchased from Fisher Scientific (Schwerte, Germany). Hydrochloric acid (37% pure) and 1 M hydrochloric acid were purchased from Grüssing GmbH (Filsum, Germany). Corn semolina was purchased from Herrnmühle (Reichelsheim, Germany), and white rice flour was purchased from Ziegler & Co. (Wunsiedel, Germany). Wheat flour type 405/550, whole grain wheat flour, wheat semolina, sucrose, glucose, and coconut fat used for the baking experiments were purchased from a local grocery store (Münster, Germany). Dulbecco’s phosphate-buffered saline (DPBS) and penicillin/streptomycin-solution (10.000 U/mL penicillin, 10 mg/mL streptomycin) were purchased from PAN™ Biotech (Aidenbach, Germany). HEPES (*N*-2-Hydroxyethylpiperazine-*N’*-2-ethane sulphonic acid) (≥99.5%) was purchased from Carl Roth. Resazurin-Na, insulin solution from bovine pancreas (10 mg/mL insulin in 25 mM HEPES), human transferrin (≥98%), human EGF (≥98%), hydrocortisone (≥98%), and sodium selenite (approx. 98%) were purchased from Sigma-Aldrich. Dulbecco’s modified eagle medium (DMEM) enriched with 4.5 g/L d-Glucose and l-Glutamine and fetal bovine serum (FBS) were purchased from Thermo Fisher Gibco (Darmstadt, Germany). Trypsin from porcine pancreas was purchased from Merck Biochrom GmbH (Berlin, Germany). For the sodium acetate buffer (50 mM): 6.8 mg/mL sodium acetate trihydrate in purified water, the pH was adjusted to 4.9 with 1 M hydrochloric acid. Resazurin working solution (440 µM): Resazurin-Na (110.5 µg/mL) in DPBS and DMSO (0.1%).

A CIT-Stock solution for the baking experiments and HPLC-MS/MS measurements was prepared by dissolving 25 mg CIT in 25 mL of ACN. The exact concentration of 1.104 mg/mL was determined photometrically using the molar extinction coefficient of 5490 mol^−1^cm^−1^ (321 nm in methanol) [3]. The concentrations of other prepared stock solutions (DCIT, **1**–**3**) were checked by UV spectroscopy using the extinction coefficients described below (“Isolation of further degradation products”) and in Brückner et al. [36].

### 4.2. Isolation of Further Degradation Products

CIT degradation starts between 100–140 °C when water is added [38,49]. To isolate further CIT degradation products in addition to DCIT [36], 4 mg CIT batches were weighed into 10 mL screw-cap glass centrifuge tubes and dissolved in 4 mL DMSO/H_2_O (95/5, *v*/*v*) [58]. Because CIT is practically insoluble in water [59], DMSO was used instead of water, and just a small amount of water was added. A total of 20.32 mg CIT was used. The reaction mixtures were heated at 140 °C for 20 min.

For subsequent removal of the DMSO, a solid phase extraction (SPE) with a polymeric reversed phase (Strata^TM^-X, 33 µm, 1 g/12 mL, Phenomenex, Aschaffenburg, Germany) was used. For this purpose, the heated reaction mixtures were combined and diluted in water (1:20). For conditioning of the column, 20 mL methanol and 20 mL H_2_O were used. The column was rinsed with 75 mL ACN, 50 mL ACN/H_2_O (80/20, *v*/*v*), and 40 mL ACN/H_2_O (50/50, *v*/*v*). The solvent of the individual fractions was removed, and the residues were resolved in 1–2 mL ACN/H_2_O (50/50, *v*/*v*).

Purification was carried out using the same semi-preparative HPLC-UV system described in Brückner et al. [36] with minor modifications as outlined below: For chromatographic separation, solvents were used without the addition of formic acid (solvent A: H_2_O, solvent B: ACN). The gradient was programmed as follows: 0 min 95% A, 1 min 95% A, 20 min 0% A, 25 min 0% A, 25.1 min 95% A, 30 min 95% A.

A chromatogram of the semi-preparative HPLC-UV is provided in the Appendix A (see Appendix A). In addition to DCIT, the largest peaks were purified and labeled as follows: *m*/*z* 395.1855 A (**1**), *m*/*z* 395.1855 B (**2**), and *m*/*z* 425.1959 (**3**). The further characterization of the substances was accomplished by using HPLC-DAD-ESI-QTOF. The yield was 2.12 mg for *m*/*z* 395.1855 A, 2.29 mg for *m*/*z* 395.1855 B, and 0.28 mg for *m*/*z* 425.1959. The available amounts were insufficient for further structure elucidation by NMR, and standards were used as quality references.

The absorption maxima and extinction coefficients were determined for all substances in ACN at the maximum absorption wavelength (*λ*_max_) and are listed below:

*m*/*z* 395.1853 A (**1**): A yellow-orange solid. UV-vis: *λ*_max_ (ACN) 295 nm; *ε*: 6612 L⋅mol^−1^⋅cm^−1^. ESI(+)HRMS: *m*/*z* 395.1864, [M+H]^+^ C_24_H_26_O_5_, calc. 395.1853 error 2.8 ppm. *m*/*z* 395.1864 (EPI^+^ 37.4 eV): *m*/*z* 378.1794 (24%), 377.1759 (100%), 359.1652 (12%), 351.1600 (17%), 349.1440 (10%), 333.1494 (5%), 325.1425 (12%), 324.1357 (27%), 323.1288 (29%), 310.1169 (7%), 309.1134 (33%), 307.1319 (22%), 306.1263 (55%), 291.1029 (17%), 268.1098 (6%), 269.1172 (5%).

*m*/*z* 395.1853 B (**2**): A yellow-orange solid. UV-vis: *λ*_max_ (ACN) 294 nm; *ε*: 4753 L⋅mol^−1^⋅cm^−1^. ESI(+)HRMS: *m*/*z* 395.1857, [M+H]^+^ C_24_H_26_O_5_, calc. 395.1853 error 1.0 ppm. *m*/*z* 395.1857 (EPI^+^ 37.4 eV): *m*/*z* 378.1774 (13%), 377.1741 (52%), 359.1634 (11%), 353.1730 (8%), 352.1653 (8%), 351.1580 (8%), 347.1271 (8%), 337.1426 (13%), 325.1403 (10%), 324.1343 (26%), 323.1269 (14%), 309.1118 (28%), 307.1319 (13%), 306.1244 (10%).

*m*/*z* 425.1959 (**3**): A yellow-orange solid. UV-vis: *λ*_max_ (ACN): 327 nm; *ε*: 4273 L⋅mol^−1^⋅cm^−1^. ESI(+)HRMS: *m*/*z* 425.1956, [M+H]^+^ C_25_H_28_O_6_, calc. 425.1959 error 0.5 ppm. *m*/*z* 425.1956 (EPI^+^ 38.1 eV): *m*/*z* 219.1018 (11%), 208.1051 (12%), 207.1018 (100%), 189.0912 (10%).

### 4.3. Model Experiments

The model experiments were carried out analogously to Brückner et al. [36]. To investigate the influence of reducing and non-reducing sugars on the degradation of CIT, the model substances α-d-glucose and d-sucrose were chosen as representatives. As a model substance for starch, methyl-α-d-glucopyranoside was used.

In each case, 20 µL of a 1 mg/mL CIT stock solution in ACN (0.08 µmol) was pipetted into brown glass vials to prevent degradation by light. Twenty µL of a 10 mg/mL α-d-glucose, d-sucrose or methyl-α-d-glucopyranoside stock solution in ACN/H_2_O (50/50, *v*/*v*) was added (1 µmol/0.6 µmol/1 µmol) and the solvent was removed at 40 °C under a stream of compressed air. Pure CIT samples served as control samples. For this purpose, 25 µL of the CIT stock solution (0.1 µmol) was pipetted into brown glass vials. The solvent was removed as described above. Sealed vials were heated at different temperatures (100; 120; 140; 160; 180 °C) for up to 1 h (10; 30; 60 min). In addition, the model compounds (1 µmol α-d-glucose, 0.6 µmol d-sucrose, or 1 µmol methyl-α-d-glucopyranoside) were heated at 160 °C without the addition of CIT (10; 20; 30; 60 min). To investigate the influence of water on the thermal degradation of pure CIT, the samples were also heated with and without the addition of water (0; 5; 50 µL water). Experiments heating CIT with model compounds were carried out in duplicate; those heating with pure CIT were in triplicate.

To confirm the formation of reaction products of CIT with reducing and non-reducing sugars, 10 µL of a ^13^C_3_-CIT stock solution (1 mg/mL in methanol) (0.04 µmol) and 10 µL of the model substance stock solution (0.5 µmol α-d-glucose; 0.3 µmol d-sucrose; 0.5 µmol methyl-α-d-glucopyranoside) were pipetted into brown glass vials, and the solvent was removed as described above. The samples were heated at 160 °C for 10 min. To confirm the participation of α-d-glucose in the formation of reaction products, CIT (0.04 µmol) and ^13^C_6_ α-d-glucose (0.5 µmol) were heated together under the same conditions.

All samples were allowed to cool after heating to room temperature, and the residues were dissolved in 500 µL H_2_O/ACN (80/20, *v*/*v*). The samples were shaken for 30 min at 350 rpm on a GFL type 3006 laboratory shaker (GFL, Gesellschaft für Labortechnik mbH, Burgwedel, Germany) to completely dissolve all degradation products formed. The samples were analyzed by HPLC-DAD-ESI-QTOF.

Furthermore, 100 mg starch was weighed into a 10 mL screw-cap glass centrifuge tube and 100 µL water, and 200 µL of the CIT-stock solution (0.8 µmol) was added, followed by gentle manual shaking for 1 min. After evaporation of the solvents from the uncapped tube overnight, the open vessel was heated for 60 min at 140 °C, allowing solvent residues and volatile reaction products to evaporate. As a control group, the experiment was repeated without the addition of CIT (blank samples). All experiments were carried out in triplicate. To be able to detect reaction products of CIT with starch by HPLC-DAD-ESI-QTOF, a digestion of the starch had to be carried out first. For this purpose, 2 mL of water and 100 µL heat-stable α-amylase from *Bacillus* sp. were added to the heated residue and shaken intensively for 30 s. The samples were heated for 30 min at 95 °C with occasional shaking. After cooling the samples to room temperature, 200 µL of the supernatant was added to 800 µL ACN to denature the α-amylase. The samples were centrifuged (15,000× *g*, 10 min, rt) (5810 R, Eppendorf AG, Hamburg, Germany) and diluted to a final concentration of 40% ACN with water, followed by HPLC-DAD-ESI-QTOF analysis.

### 4.4. Determination of the Moisture Content

For determination of the water content of the samples, as well as of the used flours, a standard drying oven method was used. First, 3–9 g of the samples were weighed into evaporating dishes, then dried for 24 h at 105 °C, and then the water content was calculated from the weight difference [34].

### 4.5. Biscuit Baking

A simple biscuit recipe according to Kuchenbuch et al. [34] was chosen to investigate the degradation of CIT during baking. The moisture content, the type of sugar, and the type of wheat flour used were varied.

To prepare the biscuit dough, flour, sugar, and melted coconut fat were combined and kneaded for 3 min (MUM4405/05, Robert Bosch Hausgeräte GmbH, München). Tap water was added, and the dough was kneaded again for 5 min. The exact quantities for preparing the doughs are summarized in Table 2. Two doughs were produced for each dough type A–F. One dough was spiked with CIT, while the other dough served as a control (blank dough). Spiking was made by adding 434.7 µL of the CIT stock solution to the tap water before mixing, resulting in a spiking level of 1000 µg CIT/kg flour (776.2–782.5 µg CIT/kg dry mass dough, Figure 8). In the case of the blank dough, the corresponding amount of ACN was added.

Two 50 g portions of the dough were shaped into discs, each with a diameter of 75 mm and a height of 10 mm, using a special mold. Subsequently, biscuits were baked on a silicon mat in an oven with circulating air (UT 6060, Heraeus, Hanau, Germany). The temperature inside the biscuit during the baking process was monitored using a temperature data logger (EL-USB-TC-LCD, Lascar Electronics, Whiteparish, United Kingdom). Two different baking conditions were investigated: low- (180 °C, 20 min) and high-temperature conditions (220 °C, 10 min). Both baking processes were repeated three times for each dough.

After baking, the biscuits were allowed to cool to room temperature and the inner part of each biscuit was cut out by a stainless-steel ring with a 52 mm inner diameter. This step was performed to remove the inhomogeneously browned areas at the edges, ensuring better comparability of the samples. The obtained inner part of the biscuit was ground (M20, IKA-Werke, Staufen, Germany) and stored at −20 °C until analysis. All CIT concentration were normalized to the dry mass. For this purpose, the water content of the used flours, as well as the water content of the biscuits after baking, was determined.

### 4.6. Sample Preparation

The method used for sample preparation is a modified method based on the official method for CIT determination in food [60]. In general, 2 ± 0.05 g sample were weighed into a 50 mL polypropylene centrifuge tube and extracted with 10 mL of ACN acidified with 1% acetic acid and 10 mL of an extraction solution consisting of H_2_O acidified with 1% acetic acid and 1.6% HCl (37%) and a sodium chloride additive of 100 g/L. The samples were shaken for 1 h at 350 rpm (GFL type 3006, Gesellschaft für Labortechnik mbH). To induce phase separation, 4 g magnesium sulphate and 1.5 g sodium chloride were added and shaken for 30 sec. The samples were then centrifuged (3200× *g*, 10 min, rt) (5810 R, Eppendorf AG). Two hundred µL of the clear supernatant were transferred to a vial, to which 790 µL ACN/H_2_O/acetic acid (40/59/1, *v*/*v*/*v*) and 10 µL of internal standard (^13^C_3_-CIT at 1 µg/mL in methanol) were added.

To ensure the accuracy of the results, a quality control sample (QC) with a known CIT content was processed and measured with every sample batch. The QC consisted of a blend of extruded samples spiked with CIT before extrusion. As the QC has a starch-rich matrix, it could also be used as a representative quality control for the sample preparation of the biscuit samples.

### 4.7. HPLC-MS/MS

Sample extracts were analyzed using an HPLC-MS/MS system consisting of a PAL HTC-xt autosampler (CTC analytics, Zwingen, Switzerland), an Agilent Infinity 1260 HPLC (Agilent Technologies, Waldbronn, Germany) in combination with an Ion Drive^TM^ Turbo V Source (SCIEX, Darmstadt, Germany), and a QTRAP 6500 triple quadrupole mass spectrometer (SCIEX). The 1260 HPLC consisted of a G4225A 1260 HiP Degasser, a G1312B 1260 Bin Pump, a G1316A 1260 thermostatted column compartment (TCC), a G1367E 1260 HiP ALS, and a G1330B 1290 Thermostat. The mass spectrometer was operated in scheduled multiple reaction monitoring (sMRM) detection mode with positive electrospray ionization (ESI), a cycle time of 800 ms, and retention time windows typically ranging from 60–210 s (with one case extending to 780 s).

A Nucleodur C18 Gravity-SB column (100 × 2 mm, 3 μm, Macherey-Nagel, Düren, Germany) equipped with a guard column with the same material (2.0 × 4 mm), in combination with ACN + 1% formic acid as solvent A and H_2_O + 1% formic acid as solvent B, was used for chromatographic separation. The linear gradient was programmed as follows: 0 min 20% A, 0.1 min 20% A, 8 min 100% A, 10 min 100% A, 10.1 min 20% A, 13 min 20% A. The flow rate was set to 0.45 mL/min, and the injection volume was 20 μL. The temperature of the column oven was set to 40 °C and the autosampler temperature to 4 °C.

The optimization of the source parameters (ion spray voltage, curtain gas, temperature, ion source gas 1 and 2) was performed by injecting a CIT-spiked extrusion-cooked sample extract several times and varying each source parameter individually (ionization voltage in 500 V steps from 2000–5000 V; curtain gas in 5 psi steps from 20–45 psi; temperature in 50 °C steps from 300–700 °C; ion source gas 1 and 2 in 5 psi steps from 45–70 psi). The optimal source parameters were determined by maximizing the intensity of the quantifier ion for CIT. Further information on the MS-method is shown in the Appendix A (Appendix A).

To determine the extent to which technological processes such as baking have an influence on the degradation of CIT and, in particular, on the formation of degradation products, the largest possible number of degradation products should be integrated into the MS method in addition to CIT. In addition to isolated standards such as DCIT and **1**–**3**, which could be used directly for tuning the collision energy (CE), declustering potential (DP), and cell exit potential (CXP), MRM-transitions of further degradation products should be integrated into the method for screening purposes. MRM-transitions for these were determined based on HPLC-DAD-ESI-QTOF data from pure CIT samples of the model experiments, which helped identify additional potentially relevant degradation products, as well as degradation products already known from the literature, namely, for example: phenol A acid (*m*/*z* 241.1084), dicitrinin A (*m*/*z* 381.1721), dicitrinin C (*m*/*z* 393.1722), and citrinin H1 (*m*/*z* 427.1779) [36]. Identified fragments of the product ion spectra could be used to determine MRM transitions. Furthermore, one sample from the model experiments (25 µg pure CIT heated for 10 min at 160 °C with the addition of 50 µL water), which contained a mix of different degradation products, was also used for tuning.

Recovery rates were determined for CIT, as well as DCIT and **1**–**3**, by spiking a certain amount of analyte to a blank sample before sample preparation. Quantification was performed using an external calibration and internal standard (^13^C_3_-labeled CIT). For the biscuit matrix, the recovery rate ranged between 95 and 112%. Limit of detection (LOD) was defined as a signal to noise ratio (S/N) of 3 (CIT: 1.1 µg/kg; DCIT: 2.6 µg/kg; **1**: 1.1 µg/kg; **2**: 1.0 µg/kg, **3**: 3.2 µg/kg) and limit of quantification (LOQ), defined as S/N of 10 (CIT: 4.4 µg/kg; DCIT: 8.6 µg/kg; **1**: 2.1 µg/kg; **2**: 2.0 µg/kg, **3**: 10.8 µg/kg).

Sample extracts analyzed for matrix-associated forms of CIT bound to starch were prepared as described under “Sample preparation for the screening of matrix-associated CIT covalently bound to starch”. The measurement using HPLC/MS-MS was also slightly modified for these samples. Instead of 20 µL, 100 µL were injected (due to the additional dilution step at the end of the sample preparation). Due to the greatly reduced number of analytes, multiple reaction monitoring (MRM) was used instead of sMRM (Appendix A).

Analyst 1.6.2 software was used for data acquisition while Sciex OS software 3.1.6 was used for data analysis for all samples.

### 4.8. Sample Preparation for the Screening of Matrix-Associated CIT Covalently Bound to Starch

One g of the biscuit samples was weighed into a 10 mL screw-cap centrifuge glass tube and suspended in 4 mL sodium acetate buffer (50 mM). Then, 200 µL of the heat-stable α-amylase from *Bacillus* sp. was added to the suspension. The samples were incubated at 95 °C for 30 min with occasional shaking. After cooling down, 100 µL of a 10 mg/mL amyloglucosidase solution in water was added, and the suspension was incubated overnight (14–16 h) at 55 °C in a drying oven (ED-115, Binder GmbH, Tuttlingen, Germany) while gently shaking (150 rpm) (GFL type 3005, Gesellschaft für Labortechnik mbH). The samples were then centrifuged (2370× *g*, 5 min, rt) (Universal 320 R, Andreas Hettich GmbH Co. KG, Tuttlingen, Germany), and 200 µL of the supernatant was diluted in 800 µL ACN. After the samples were centrifuged again (15,000× *g*, 10 min, rt) (5810 R, Eppendorf AG), 800 µL H_2_O were added to 200 µL of the supernatant. The samples were measured using HPLC-MS/MS (see section “HPLC-MS/MS”).

### 4.9. Sample Preparation for the Screening of Matrix-Associated CIT Covalently Bound to Protein

One g of the ground biscuit samples was weighed into a 10 mL screw-cap centrifuge glass tube and suspended in 4 mL H_2_O. Two hundred µL α-amylase were added, and the samples were incubated at 95 °C for 30 min with manual occasional shaking. The samples were then centrifuged (2370× *g*, 5 min, rt) (Universal 320 R, Andreas Hettich GmbH), and the supernatant was discarded. Five hundred µL of pronase solution (6.5 mg/mL in PBS-buffer) and 500 µL of PBS buffer were added. The samples were incubated overnight at 37 °C (ED-115, Binder GmbH) while gently shaking (200 rpm, GFL type 3005, Gesellschaft für Labortechnik mbH) and then centrifuged again (2370× *g*, 5 min, rt) (Universal 320 R, Andreas Hettich GmbH). The supernatant was further purified using a solid phase extraction (SPE) cartridge (Oasis MAX 1 mL, 30 mg, Waters, Eschborn, Germany) analogous to the protocol already described [36]. The samples were then measured with HPLC-DAD-ESI-QTOF and checked for the previously published features (reaction products of CIT and gluten).

### 4.10. Exact Mass Measurements with HPLC-DAD-ESI-QTOF

The exact mass measurements with HPLC-DAD-ESI-QTOF were carried out as described in Brückner et al. [36].

### 4.11. Data Processing and Statistical Analysis

Data processing and statistical analysis of the HPLC-DAD-ESI-QTOF data was carried as described in Brückner et al. [36]. In addition, Compass Data Analysis 6.1 and MetaboScape 2024b (Bruker Daltonik, Bremen, Germany) were used for data processing.

### 4.12. Cell Culture

Immortalized human kidney epithelial-cells (IHKE-cells, provided by S. Mollerup, National Institute of Occupational Health, Norway) [61] were cultivated under cell culture conditions (37 °C, 5% CO_2_) in DMEM/F-12 medium supplemented with HEPES (15 mM), penicillin (100,000 U/L), streptomycin (100 mg/L), fetal bovine serum (1%), human transferrin (5 mg/L), human insulin (5 mg/L), Na-selenite (5 µg/L), EGF (10 µg/L), and hydrocortisone (500 µg/L). Cells were subcultured twice a week at a ratio of 1:30. Cells were used in the passages 167–170.

### 4.13. Cytotoxicity Analysis via Resazurin Assay

The cytotoxicity of DCIT compared to CIT was assessed by determining the cell viability using the resazurin assay, as described by Rottkord et al. [61]. Further details are provided in the Appendix A.

## Figures and Tables

**Figure 1 toxins-17-00086-f001:**
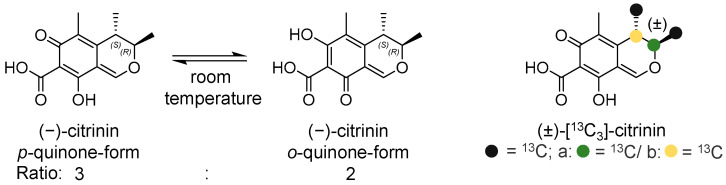
Chemical structure of CIT in its tautomeric forms *p*-quinone and *o*-quinone (3:2 equilibrium between the *p*- and *o*-tautomers in the solid state at room temperature), as well as the chemical structure of the ^13^C_3_-labeled CIT.

**Figure 2 toxins-17-00086-f002:**
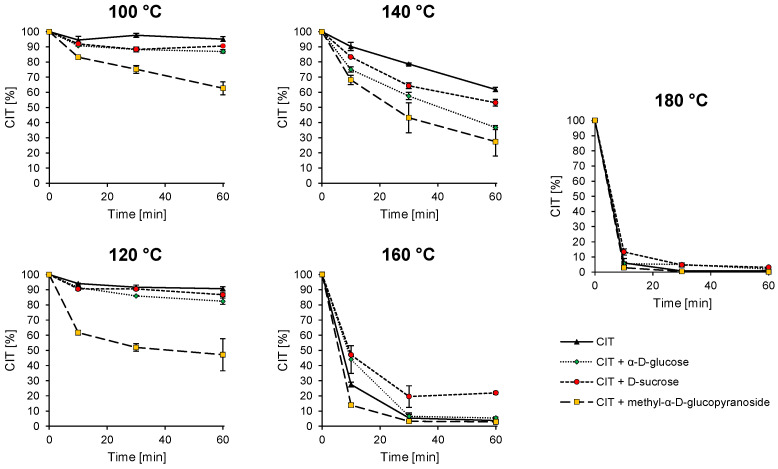
Degradation curves representing the thermal stability of CIT during heating for different times (10, 30, 60 min) and temperatures (100, 120, 140, 160, 180 °C) with and without the addition of model compounds to mimic different carbohydrates (α-d-glucose; d-sucrose; methyl-α-d-glucopyranoside). Methyl-α-d-glucopyranoside was used as a model compound for starch, reducing sugars were simulated by α-d-glucose, and non-reducing sugars and disaccharides were simulated by d-sucrose.

**Figure 3 toxins-17-00086-f003:**
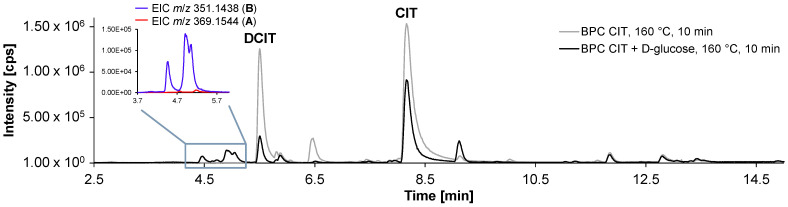
Base peak chromatograms (BPC) of CIT heated with α-d-glucose (black trace) at 160 °C for 10 min and of pure CIT heated under the same conditions (grey trace). When comparing the two BPCs, it is noticeable that new peaks at a retention time window between 4.5–5.7 min are formed only when α-d-glucose was added. The enlargement shows the extracted ion chromatograms (EIC) for *m*/*z* 351.1438 and 369.1544.

**Figure 4 toxins-17-00086-f004:**
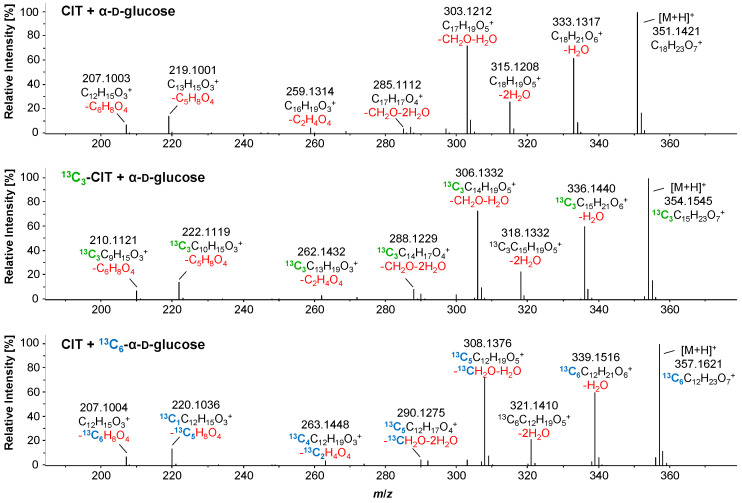
Top: HRMS product ion spectrum of the reaction products **B** of CIT with α-D-glucose with *m*/*z* 351.1445, retention time: 5.04 min, CE 21.3 eV ([M+H]^+^, C_18_H_22_O_7_). Neutral losses are shown in red under the respective sum formula, calculated from the exact mass. Middle: HRMS product ion spectrum of the reaction product of ^13^C_3_-CIT with α-D-glucose with *m*/*z* 354.1545, retention time: 5.00 min, CE 21.4 eV ([M+H]^+^, ^13^C_3_C_15_H_22_O_7_). In addition to the neutral losses shown in red, the ^13^C-labelling of CIT is indicated in green. Bottom: HRMS product ion spectrum of the reaction product of CIT with ^13^C_6_-α-D-glucose with *m*/*z* 357.1621, retention time: 5.01 min, CE 21.4 eV ([M+H]^+^, ^13^C_6_C_12_H_22_O_7_). In addition to the neutral losses shown in red, the ^13^C-labelling of α-D-glucose is indicated in blue. By comparing the different fragment spectra, conclusions can be drawn regarding which fragments can be assigned to CIT and which to α-D-glucose.

**Figure 5 toxins-17-00086-f005:**
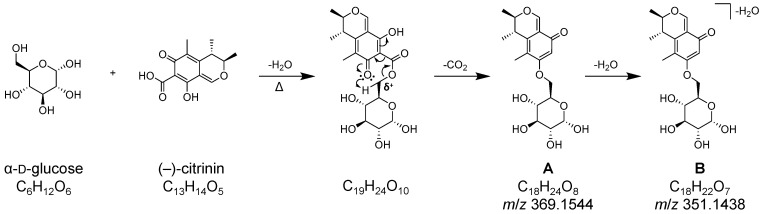
Postulated reaction pathway for the formation of the reaction products with the sum formulas C_18_H_24_O_8_ (**A**) and C_18_H_22_O_7_ (**B**), which are formed when CIT and α-d-glucose are heated together. The first step in the postulated reaction is an ester formation, followed by cyclisation and the exclusion of CO_2_. In reality, a variety of isomers are formed, with only one possible isomer shown here as an example.

**Figure 6 toxins-17-00086-f006:**
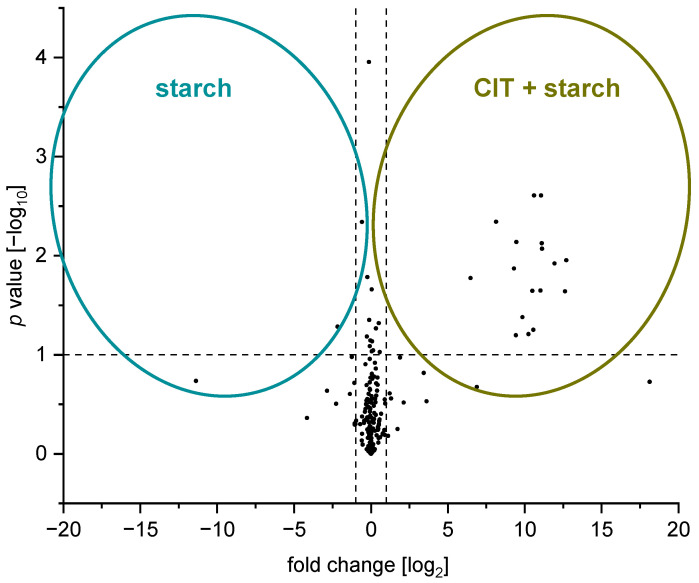
Visualization of the univariate statistical analysis (fold-change analysis with combined *t*-test analysis) using a volcano plot. Significant features (significance level of 90%) of the test group (CIT heated with starch), i.e., features that differ statistically significantly from the control group appear on the right side of the plot. Significant features (significance level of 90%) of the control group (starch without the addition of CIT) are shown on the left side of the plot. Samples were digested with α-amylase before measurement with HPLC-DAD-ESI-QTOF.

**Figure 7 toxins-17-00086-f007:**
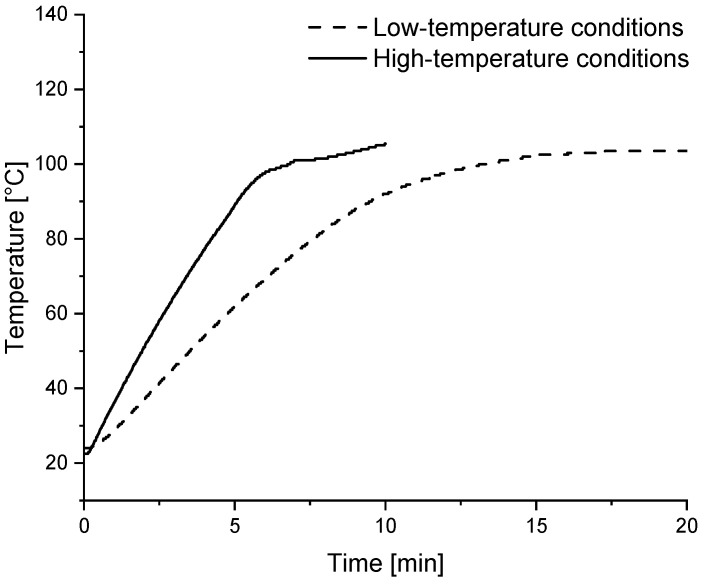
Temperature curves recorded inside the biscuits during the low- (180 °C, 20 min) and high-temperature (220 °C, 10 min) baking of dough C.

**Figure 8 toxins-17-00086-f008:**
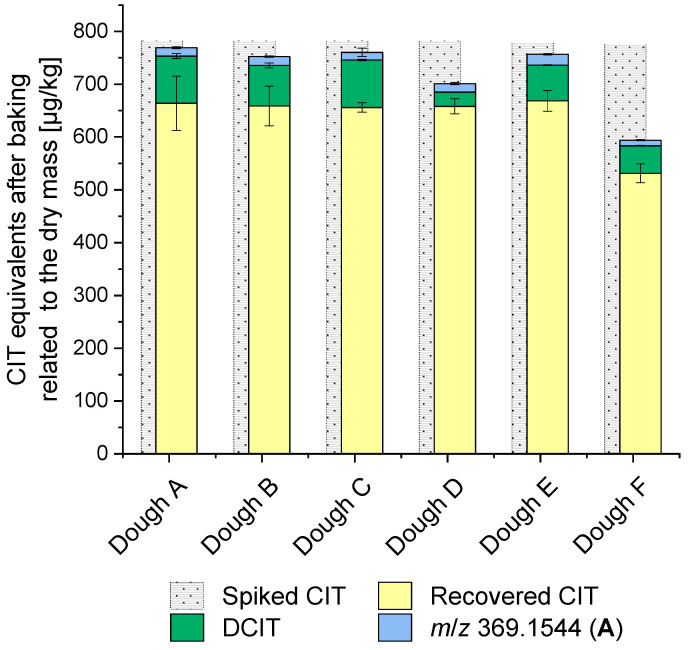
CIT degradation during biscuit-making in the different doughs (dough A–C: variation of the water content; dough C–D: variation of the sugar used; dough C, E, and F: variation of the wheat flour type used (Table 2). All doughs were spiked with 776.2–782.5 µg CIT/kg dry mass dough.) In addition to the recovered CIT, the amount of DCIT formed was quantified. The starch-bound CIT (reaction products **A**) was semi-quantified using the CIT calibration due to the absence of an analytical standard. For the analysis of the covalently bound CIT to starch (**A**), only biscuits baked under low-temperature conditions (180 °C, 20 min) were used. All levels are reported in µg CIT equivalent/kg dough, related to the dry mass.

**Figure 9 toxins-17-00086-f009:**
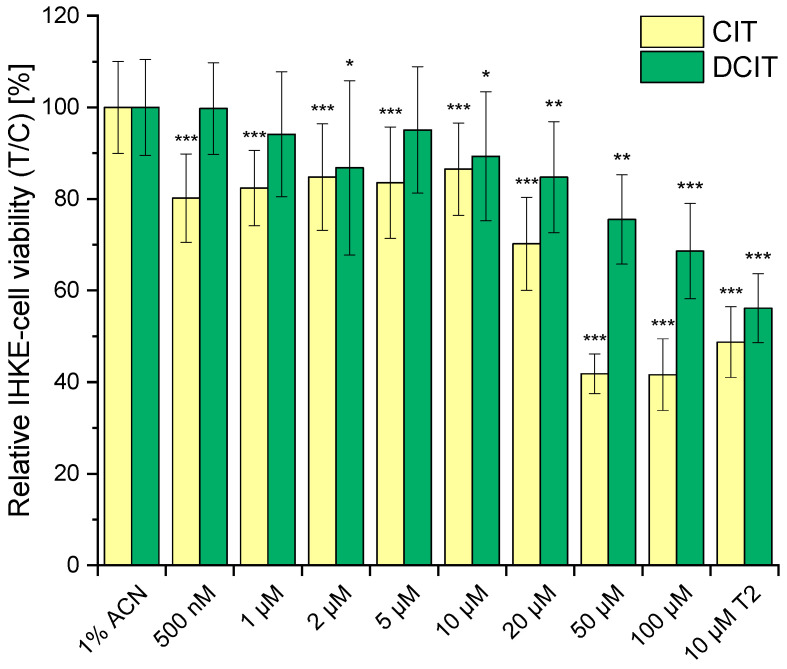
Cellular viability of IHKE cells was assessed after 24 h incubation with 500 nM to 100 µM CIT and DCIT using the resazurin assay. Results, presented after blank subtraction, were normalized to the negative control (1% acetonitrile). As a positive control, 10 µM T-2 toxin was used. Cytotoxicity data for each cell viability tested was statistically evaluated using an unpaired, heteroscedastic Student’s *t*-test relative to the 1% ACN negative control (*n* = 6 × 3; * *p* ≤ 0.05, ** *p* ≤ 0.01, *** *p* ≤ 0.001).

**Table 1 toxins-17-00086-t001:** Features likely to be reaction products of CIT and starch, analyzed by HPLC-DAD-ESI-QTOF after digestion with α-amylase, are presented. For each feature, the mass-to-charge ratios (*m*/*z*), retention times, their maximum intensity, the tentatively assigned sum formula, the difference between calculated and determined *m*/*z* (Δppm), the sum formula change compared to CIT (C_13_H_14_O_5_), and the -log_10_(*p*-value) and log_2_(fold change) are provided.

m/z	Retention Time [min]	Max. Intensity [Counts]	Assigned Sum Formula	Sum Formula Change	Δppm	*p*-Value [−log_10_]	Fold Change [log_2_]
369.1548	4.95	1.50 × 10^6^	C_18_H_24_O_8_	+C_5_H_10_O_3_	1.0	2.14	9.45
531.2080	4.85	6.90 × 10^5^	C_24_H_34_O_13_	+C_11_H_20_O_8_	1.4	1.95	12.7
369.1546	5.16	4.25 × 10^5^	C_18_H_24_O_8_	+C_5_H_10_O_3_	0.5	1.65	11.0
351.1440	5.07	2.01 × 10^5^	C_18_H_22_O_7_	+C_5_H_8_O_2_	0.4	2.07	11.1
693.2749	4.73	7.96 × 10^4^	C_30_H_44_O_18_	+C_17_H_30_O_13_	1.5	1.2	9.4

**Table 2 toxins-17-00086-t002:** Recipe for the different doughs A–F prepared. Variation of the water content (dough A–B, C), variation of the used sugar (dough C–D), and variation of the wheat flour type (dough C, E, and F).

Ingredient	Dough A	Dough B	Dough C	Dough D	Dough E	Dough F
	Variation in Water Content		Variation of the Sugar		Variation of the Wheat Flour Type	
	Low	High	Sucrose	Glucose	Type 550	Whole Grain
Flour [g]	480 ^1^	480 ^1^	480 ^1^	480 ^1^	480	480
Water [g]	110	165	137	137	140	143
Coconut Fat [g]	80	80	80	80	80	80
Sugar [g]	120 ^2^	120 ^2^	120	120	120 ^2^	120 ^2^
Absolute water content ^3^ [%]	22.5	27.5	25.0	25.0	25.0	25.0

^1^ Wheat flour type 405 was used as flour. ^2^ Sucrose was used as sugar. ^3^ Water content of the flour considered.

## Data Availability

The original contributions presented in this study are included in the article/Appendix A. Further inquiries can be directed to the corresponding authors.

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
