# Peer review of "Thermal Stability and Matrix Binding of Citrinin in the Thermal Processing of Starch-Rich Foods"

_toxins, 2025, doi:10.3390/toxins17020086_

Round 1
Reviewer 1 Report
Comments and Suggestions for Authors
This research focuses on the thermal stability and matrix-binding characteristics of citrinin (CIT) during the thermal processing of starchy foods. Through model experiments and actual baking experiments, combined with multiple analytical techniques, it delves deeply into the reactions between CIT and carbohydrates, as well as the degradation products. This study makes a significant contribution to understanding the change mechanisms of CIT during food processing. The experimental design is relatively reasonable, and the data is abundant.
1.In the results section, the labels and explanations of some charts and graphs (such as the 211 volcano plot) could be more detailed so that readers can understand the information reflected by the data more intuitively. The chemical formulas in line 180 are not aligned. It is also necessary to consider whether the units need to be uniformly labeled.
2..A preliminary comparison of the cytotoxicity of DCIT and CIT in IHKE cells was conducted. However, no in-depth research was carried out on the toxicity of carbohydrate-bound CIT and its intestinal metabolism. An experimental group with only heated carbohydrates as the control could be added.
3.In some chapters, the wording is relatively complex, and the language can be simplified appropriately to enhance the readability of the paper. For example, in the Results and Discussion section, the description and analysis of some experimental data can highlight the key points more concisely and clearly. In addition, the labeling and explanation of charts can be further improved, enabling readers to quickly and accurately understand the information conveyed by the charts. In terms of citation, some recent relevant research results can be appropriately added to further highlight the status and contribution of this study in the field. The paper cites 41 references, which might be relatively scarce. Additionally, there are few references from the past five years. It is recommended to consult more literature, theoretical research, and conclusions from the past three years to ensure that the content presented is at the forefront of the field.
4.I think the reaction mechanism is insufficient. Although a possible pathway for the reaction between CIT and α-D-glucose was proposed, some steps are still based on speculation, lacking more direct evidence for support.
5.Reaction mechanism details: In the reaction mechanism between CIT and carbohydrates, key steps lack support. It is suggested to supplement more experimental data and further explore the reasons for the differences in reaction activity between different carbohydrates and CIT, in order to reveal the essential laws of the reaction.
6.When discussing the impact of dough F on CIT degradation in cookie baking experiments, in addition to mentioning the compositional differences of whole wheat flour, further exploration can be conducted on the specific chemical reactions or physical mechanisms that these components (such as dietary fiber, minerals, etc.) may participate in, as well as how they synergistically affect CIT degradation and binding
7.Improve control settings: In model experiments studying the reaction between CIT and different carbohydrates, in addition to setting a control without model compounds, it is possible to consider adding a control that only heats carbohydrates to eliminate the interference signals that carbohydrates may generate during the heating process, further enhancing the credibility of the experimental results.
8.The results of the cytotoxicity test 5 μM and 10 μM have different trends than the overall and should be interpreted.
Reviewer 2 Report
Comments and Suggestions for Authors
In this manuscript, the authors investigate the thermal stability of citrinin and its ability to bind to food substrates, especially starch. The manuscript is interesting, however, more experimental characterization and data should be provided.
1. In which foods is CIT toxin contamination common? Is contamination in cereals enough to warrant attention? More detailed data and references are needed to explain this concern.
2. The data obtained from the reaction experiment of citrinin with methyl-α-D-glucopyranoside, α-D-glucose and D-sucrose are very superficial phenomena. The authors need to conduct necessary characterization of the effect or combination of citrinin with these three different carbohydrates to verify the authors' hypothesis.
3. In the degradation experiment of CIT in baked products, two different temperatures were set, and the duration of the final temperature was inconsistent. The multi-dimensional differences in experimental factors may not be conducive to the interpretation of the test results, because lower heating for a longer period of time may reach a cumulative effect. Therefore, it should be considered to control the same heating time at different temperatures.
4. In Figure 7, it can be observed that the recovery of CIT is significantly reduced. Please try to explain why this is and how this is related to the different types of dough?
5. The authors only considered the potential nephrotoxicity of DCIT, but from the manuscript, CIT has multiple products after thermal processing. What are the toxicities of other products? What is the combined toxicity of these products? It is worth further consideration.
Author Response
see enclosed file

Reviewer 3 Report
Comments and Suggestions for Authors
The authors have presented a clear, concise and well planned experiment to elucidate the extent of citrinin binding to carbohydrates as well as the potential products that are produced by this action. This work adds value to the literature of citrinin as well as other similar compounds.
One major point I have is related to ‘Reaction Product B’. This reaction product of citrinin is purported throughout to be a single product. Firstly, it is referred to as ‘Reaction Product B’ throughout and not ‘reaction products’. In the zoom in of the chromatogram in Figure 3, there are atleast 3 isobaric peaks present, suggesting 3 products.
I believe the authors need to address this in the results section and discuss it in the discussion section. This is especially relevant for the figure showing a potential reaction mechanism leading to a single product.
Another issue that I believe doesn’t line up and should be discussed is that the main reaction product found in the reactions with the more simple saccharides is the m/z 351 product, not the m/z 369 product. However during the baking with biscuits, I believe that the m/z 351 product was not detected while the m/z 369 was? If this is the case, can a more clear explanation of this be provided or put forth?
Minor comments
Line 27 ‘und’ should be and
Figure 1 à is anything known about the equilibrium constant of the p- and o-quinone form that could be added to this figure?
Something that was not discussed in the introduction is why we would like to understand the nature of citrinin binding to matrix material which may be that there is a potential for the release of the bound toxin. This was mentioned as a possibility with T-2 and HT-2 on L60, but is there any information to suggest that bound forms of citrinin could be getting released during digestion?
Figure 1. I find the data for glucose and methyl-glucopyranoside difficult to understand. Glucose also has the same types of hydroxyl groups as methyl-glucopyranoside in addition to being a reducing sugar. Why would citrinin be more stable with glucose?
Figure 3, why are certain m/z values in boxes?
L188: As m/z 513 was not detected in positive mode, was the analysis performed in negative mode? In addition to the [M-H]- saccharides can form adducts with Cl- and HAc- as well.
Throughout the work, the ‘intensity’ term is used to approximate the amount of the various reaction products. Why was intensity/cps used in place of peak area?
L331. I am unsure what ‘due to a missing reference,’ means. Is that to say there was no analytical standard?
Author Response
see enclosed file

Round 2
Reviewer 2 Report
Comments and Suggestions for Authors
The authors carefully considered and responded to the reviewers' comments and suggestions, and the quality of the manuscript has been greatly improved. However, how citrinin specifically binds to methyl-α-D-glucopyranoside, α-D-glucose, and D-sucrose is still worthy of attention.
Comments on the Quality of English Language
As a reviewer from a non English speaking country, I do not have the authority to evaluate the English expression quality.
Author Response
see enclosed file
